# Foundation Models for Partial Causal Identification

**Alexis Bellot** [1]    **Anish Dhir** [2]

## Abstract

This paper investigates the development of causal foundation models for bounding the effect of interventions and counterfactuals from observational data. We show that a canonical prior can be defined with full support over the space of structural causal models with discrete observables. With this canonical prior, we translate the problem of bounding counterfactuals into that of learning distributions over functions that map data (and possibly structural assumptions) to a causal query of interest. This extends the promising causal foundational modelling paradigm to the estimation of *partially-identifiable* causal effects, i.e., under unobserved confounding, where multiple values are equally compatible with the observed data and prior structural assumptions.

## 1. Introduction

In the background of any investigation into causal inference are concerns about *identifiability*: roughly, that the causal query may not be uniquely computable from the available data even in the large sample limit. It is commonly acknowledged that a relatively detailed understanding of the data-generating process is necessary for identification (Pearl, 2009; Tian & Pearl, 2002; Bareinboim et al., 2022).

In practice, it is natural to ask for results and methods that are more "universally" applicable and do not require having access to qualitative knowledge of the system's structure. In cases where the query is not uniquely computable we might ask: what is the tightest possible bound that contains all values consistent with the observed data and additional assumptions we might pose? This line of research falls under the rubric of *partial causal identification* (Manski, 1990; Balke & Pearl, 1994; Tian & Pearl, 2000; Finkelstein & Shpitser, 2020; Zhang & Bareinboim, 2021b; Zhang et al., 2022; Sachs et al., 2023; Duarte et al., 2024; Jung & Kang,

2026). This generalizes the causal estimation problem to the identification of a set and has found applications in econometrics (Tamer, 2010), epidemiology (Mullahy et al., 2021), and AI (Jalaldoust et al., 2024; Li et al., 2025; Joshi et al., 2024; Bellot et al., 2023).

Despite considerable progress, the existing toolbox for partial identification is highly fragmented. Each instance of the problem has historically required a bespoke derivation: specific causal structures, queries, and types of input datasets (e.g. observational vs interventional), require separate optimization runs. On the other hand, Prior-Data Fitted Networks (Garnelo et al., 2018; Müller et al., 2022) are establishing a powerful paradigm for learning distributions over functions between two general spaces, such as the space of datasets and causal queries. This idea has led to the development of causal foundation models (Robertson et al., 2025; Balazadeh Meresht et al., 2025; Dhir et al., 2025; Reuter et al., 2026; Ma et al., 2025; Bynum et al., 2025), mapping data to the value of a causal quantity by training on pairs of synthetically-generated examples. Most causal foundation models operate exclusively in settings where the query of interest is *point-identifiable*. Dhir et al. (2025); Reuter et al. (2026) apply this idea to partially identified regimes, producing a posterior distribution that does not necessarily concentrate around a single value. However, one cannot necessarily derive valid bounds from their posteriors in a principled manner because their prior is not guaranteed to give mass to *all* values of the query consistent with the data.

In this paper, we develop a *foundation model for partial identification* that, given an observational dataset, outputs a posterior distribution over a causal query whose support provably encodes the identified set. Our distinctive contribution is the definition of a *canonical prior* with full support over *all* SCMs with discretely-valued observables. This enables us to derive bounds on any causal query from any observational dataset (and possibly other prior knowledge, like a causal diagram) that are guaranteed to be valid and tight asymptotically as the dataset size grows.

### 1.1. Preliminaries

The basic framework of our analysis rests on *structural causal models* (SCMs) (Pearl, 2009, Def. 7.1.1), which formalize the notion of a data-generating process. In addition to specifying the distribution of data, these models also

---

[1] Independent [2] University College London. Correspondence to: Alexis Bellot <abellotolivan@gmail.com>.

*Proceedings of the 2nd ICML Workshop on Foundation Models for Structured Data*, Seoul, South Korea. 2026. Copyright 2026 by the author(s).

specify the generative mechanisms that produce them. For the purpose of causal inference and learning, SCMs provide a broad, fine-grained hypothesis space.

An SCM $M$ is a tuple $\langle \boldsymbol{V}, \boldsymbol{U}, \mathcal{F}, P \rangle$ where $\boldsymbol{V}$ is a set of endogenous variables and $\boldsymbol{U}$ is a set of exogenous variables. $\mathcal{F}$ is a set of functions where each $f_V \in \mathcal{F}$ decides values of an endogenous variable $V \in \boldsymbol{V}$ taking as argument a combination of other variables in the system. That is, $V \leftarrow f_V(\mathbf{Pa}_V, \boldsymbol{U}_V)$ where observed parents $\mathbf{Pa}_V \subseteq \boldsymbol{V}$ and unobserved parents $\boldsymbol{U}_V \subseteq \boldsymbol{U}$. An intervention $do(x)$ replaces the causal mechanism $f_X$ with the assignment $X \leftarrow x$ corresponding to a sub-model $\mathcal{M}_x$. We will use the counterfactual notation $Y_x(\boldsymbol{u})$ to denote the value of the outcome $Y$ in $\mathcal{M}_x$ given $\boldsymbol{U} = \boldsymbol{u}$. Drawing values of exogenous variables $\boldsymbol{U}$ following the distribution $P(\boldsymbol{U})$ induces a counterfactual variable $Y_x$. Specifically, the event $\boldsymbol{Y_x} = \boldsymbol{y}$ (for short, $\boldsymbol{y_x}$) can be read as "$\boldsymbol{Y}$ would be $\boldsymbol{y}$ had $\boldsymbol{X}$ been $\boldsymbol{x}$".

With a given SCM $M$, for subsets $Y, \ldots, \boldsymbol{Z}, \boldsymbol{X}, \ldots, \boldsymbol{W} \subseteq \boldsymbol{V}$, the distribution over counterfactuals $\boldsymbol{Y_x}, \ldots, \boldsymbol{Z_w}$ is defined as:

$$P_M(\boldsymbol{y_x}, \ldots, \boldsymbol{w_z}) = \tag{1}$$
$$\int \mathbb{1}\{\boldsymbol{Y_x}(\boldsymbol{u}) = \boldsymbol{y}, \ldots, \boldsymbol{W_z}(\boldsymbol{u}) = \boldsymbol{w}\}\, dP(\boldsymbol{u}).$$

Given an observational distributions, the best bound on a counterfactual query is defined as follows.

**Definition 1** (Optimal Counterfactual Bound). For a distribution $P(\boldsymbol{V})$, the optimal bound $[l, u]$ over a counterfactual probability $P(\boldsymbol{y_x}, \ldots, \boldsymbol{z_w})$ is defined as, respectively,

$$\min / \max_{M \in \mathbb{M}} \quad P_M(\boldsymbol{y_x}, \ldots, \boldsymbol{z_w}) \tag{2}$$
$$\text{s.t.} \quad P_M(\boldsymbol{V}) = P(\boldsymbol{V}),$$

where $P_M$ denotes the distribution induced by the SCM $M$.

The problem formulation is presented in the language of counterfactuals to encompass any causal or counterfactual statement of interest, such as a causal effect $P(\boldsymbol{y_x})$[1]. We will write $\mathcal{I}_\Psi(P_M)$ or just $\mathcal{I}$ for the true identified set.

Each SCM $M$ can also be associated with a *causal diagram* $\mathcal{G}$, where nodes represent endogenous variables $\boldsymbol{V}$, directed edges represent the arguments $\mathbf{Pa}_V$ of each function $f_V$, and bi-directed edges represent the presence of a common latent variable in the arguments of two functions. We will leverage a special type of clustering of nodes in $\mathcal{G}$ called the *confounded component* (or c-component for short) from (Tian & Pearl, 2002). For a causal graph $\mathcal{G}$, a subset $\boldsymbol{C} \subseteq \boldsymbol{V}$

is a c-component if any pair $V_i, V_j \in \boldsymbol{C}$ is connected by a bi-directed path in $\mathcal{G}$. Given a diagram $\mathcal{G}$, we write $\mathbb{C}(U)$ for the c-component of $U$.

Throughout this paper, we assume that domains of endogenous variables $\boldsymbol{V}$ are **discrete and finite**; exogenous variables $\boldsymbol{U}$ could a priori take values in any (continuous) domain. The counterfactual probability $P(\boldsymbol{Y_x}, \ldots, \boldsymbol{Z_w})$ defined in Eq. (1) is thus a categorical distribution. We use $\text{supp}_V$ to denote the support of the distribution of $V \in \boldsymbol{V}$.

## 2. Universal Causal Estimation

We work towards the development of a *universal* solver for partial identification. Rather than targeting a specific causal diagram or a specific bound algorithm, a trained model implicitly solves the optimal bounding problem for any query derived from the SCM.

The idea is to bound the value of a counterfactual query $\Psi := P(\boldsymbol{y_x}, \ldots, \boldsymbol{w_z})$ given an observational dataset $\mathcal{D} \sim P(\boldsymbol{V})$ by examining the pushforward induced by the posterior over SCMs. For any measurable set $A \subseteq [0, 1]$ define

$$\Pi(\Psi \in A \mid \mathcal{D}) = \int \mathbb{1}\{\Psi_M \in A\}\, d\Pi(M \mid \mathcal{D}). \tag{3}$$

Optimal bounds could then be approximated by the posterior support bounds,

$$\inf / \sup\ \text{supp}\, \Pi(\Psi \mid \mathcal{D}). \tag{4}$$

These correspond to the 0- and 1-quantile limits. We use $\mathcal{I}_{\Pi(\cdot|\mathcal{D})}$ to denote the set of values defined by the posterior support. The shape of the posterior may depend on the prior, but $\mathcal{I}_{\Pi(\cdot|\mathcal{D})}$ should capture the full range of counterfactual values compatible with the observed data if priors have full support as the dataset size grows.

Following the paradigm of causal foundation models, we propose to learn a parametric approximation to the posterior directly from synthetic examples sampled from a prior over the space of SCMs $\Pi(M), M \in \mathbb{M}$.

### 2.1. Canonical Priors

In general systems, the set of functions $\{f_V : V \in \boldsymbol{V}\}$ between variables and probability measures $P(\boldsymbol{U})$ that define an SCM $M$ may be arbitrary. This makes it difficult to define good priors.

For discrete observables $\boldsymbol{V}$, however, any set of counterfactuals derived from an SCM $M$ can be equivalently obtained from a canonical SCM $N$ parameterized by categorical probability mass functions $P(U), U \in \boldsymbol{U}$ with a finite and fixed cardinality $|\text{supp}_U|, U \in \boldsymbol{U}$. This parameterization is known as the family of *canonical* SCMs proposed in (Zhang et al., 2022, Def. 2.3).

---

[1]This definition can be extended to a collection of observational and experimental distributions, and additional constraints over the space of SCMs $\mathbb{M}$.

**Definition 2** (Canonical SCM). A canonical SCM $M$ is an SCM parameterized by categorical probability mass functions $P(U), U \in \boldsymbol{U}$ with a finite and fixed cardinality,

$$|\mathrm{supp}_U| = \prod_{V \in \mathbb{C}(U)} |\mathrm{supp}_{PA_V} \mapsto \mathrm{supp}_V|. \qquad (5)$$

$|\mathrm{supp}_{PA_V} \mapsto \mathrm{supp}_V|$ denotes the number of functions from the (discretely-valued) parents of $V$ to the values of $V$. $\mathbb{C}(U)$ denotes the $c$-component of $U$.

For example, let $\mathcal{G} : \{X \to Y, X \leftrightarrow Y\}$ with binary $X, Y \in \{0, 1\}$. An SCM corresponding to this structure parameterized by a latent variable $U$ with cardinality $|\mathrm{supp}_U| = |\mathrm{supp}_X| \times |\mathrm{supp}_X \mapsto \mathrm{supp}_Y| = 2 \times 4 = 8$ is a canonical SCM.

The canonical parameterization is maximally expressive, that is, it can represent *any* SCM $M$ with discrete observables $\boldsymbol{V}$ (Zhang et al., 2022, Thm. 2.4). This suggests a natural family of priors over SCMs that we call *canonical priors*.

**Definition 3** (Canonical Prior). Given a set of observables $\boldsymbol{V}$, a canonical prior $\Pi_0$ is a probability measure on the joint space of causal diagrams $\mathcal{G}$ and the parameter space $\boldsymbol{\theta} = \{P(u) : u \in \mathrm{supp}_U, U \in \boldsymbol{U}\}$ of the corresponding family of canonical SCMs.

We consider a particular instance of the canonical prior that uses the Erdős-Rényi model for the causal diagram and the Dirichlet distribution for the exogenous parameters, as follows:

$$\Pi_0(\mathcal{G}) = \mathrm{ErdősRényi}(\mathcal{G}; q_D, q_B), \qquad (6)$$
$$\Pi_0(\boldsymbol{\theta} \mid \mathcal{G}) = \prod_{U \in \boldsymbol{U}} \mathrm{Dir}(\boldsymbol{\theta}_U; \alpha\boldsymbol{1}).$$

$q_D, q_B \in (0, 1)$ are the probabilities of including a directed and a bidirected edge, respectively, $\alpha > 0$ is a concentration parameter and $\boldsymbol{\theta}_U \in \Delta_{|\mathrm{supp}_U|-1}$ is the exogenous distribution for $U$.

Under this prior, every graph receives positive probability from the Bernoulli edge model and every categorical distribution $\boldsymbol{\theta}_U := \{P(u) : u \in \mathrm{supp}_U\}, U \in \boldsymbol{U}$ receives positive probability from the Dirichlet distribution.

**Proposition 1** (Full Support). *The canonical prior $\Pi_0$ in Eq. (6) has full support over $\mathbb{M}$: for any SCM $M \in \mathbb{M}$ and any open neighborhood $\mathcal{U}$ of $M$ in $\mathbb{M}$, $\Pi_0(\mathcal{U}) > 0$.*

Proofs are given in Appendix B. The key implication of Prop. 1 is that with full support canonical priors we can guarantee that the posterior $\Pi_0(\cdot \mid \mathcal{D})$ remains faithful to the data in the sense that no SCM consistent with the observed distribution is assigned zero mass a priori.

## 2.2. Prior-Data Fitted Networks

To approximate this posterior distribution for arbitrary queries and data distributions we use a Prior-Data Fitted Network (PFN) (Müller et al., 2022). They are a class of neural processes (Garnelo et al., 2018) that meta-learn to approximate the Bayesian posterior predictive distribution with a parametric model $q_\omega(\Psi \mid \mathcal{D})$ directly from samples from the prior.

Following (Robertson et al., 2025; Dhir et al., 2025; Reuter et al., 2026), we adopt a transformer-based architecture with input embeddings designed to preserve variable values, and types (e.g., interventions, observations) as well as correlations between them. We then stack multiple transformer blocks with attention mechanisms that alternate at each layer between sample-wise attention, which shares information across samples, and feature-wise attention, which models dependencies among variables within each sample. The counterfactual query $\Psi$ is then decoded from the resulting representation through a linear head.

The model $q_\omega$ is then trained to predict a discrete distribution over the possible binned values of $\Psi \in [0, 1]$ given the observational dataset $\mathcal{D}_n$. The training data is given by batches of examples $(\mathcal{D}_n, \Psi)$ generated by repeatedly drawing canonical SCMs $M \sim \Pi_0$ from the prior using Eq. (6), generating an observational dataset $\mathcal{D}_n \sim P_M$ by sampling $n$ i.i.d. observations, and computing the ground-truth counterfactual value $\Psi(M)$ from $M$. Architectural and training details are given in Appendix D.

## 2.3. Extensions

Existing partial identification methods exploit additional qualitative knowledge about the underlying data generating process, for example encoded in a causal diagram (Zhang et al., 2022) or equivalence class (Bellot, 2023). One can readily condition on a partial or full specification of the causal diagram $\mathcal{G}$ by training the PFN on tuples $(\mathcal{G}, \mathcal{D}_n, \Psi)$ derived from the SCM $M$. This has been shown to be tractable by (Reuter et al., 2026).

## 3. What do we converge to?

The following result shows that the PFN's predictive distribution converges, as the dataset grows, to the posterior induced by the canonical prior conditional on the observational distribution.

**Theorem 1** (Consistency). *Let $\mathcal{D}_n$ be a dataset of $n$ i.i.d. observations generated from an SCM $M$. As $n \to \infty$, the predictive distribution $q_\omega(\Psi \mid \mathcal{D}_n)$ of the trained PFN converges weakly to the conditional prior:*

$$q_\omega(\Psi \mid \mathcal{D}_n) \to \Pi_0(\Psi \mid P_M), \qquad (7)$$

*where $\Pi_0(\Psi \mid P_M)$ is the prior distribution restricted to the*

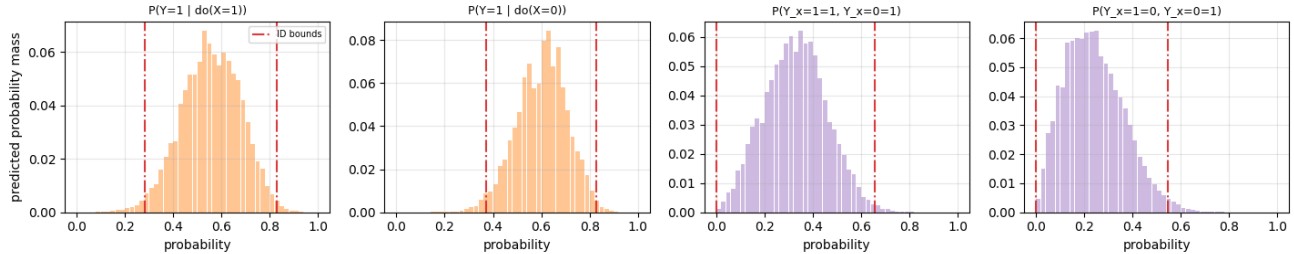

*Figure 1.* Predicted posterior distributions for interventional and counterfactual probabilities in a two-variable binary SCM experiment with 100 data samples. Dashed vertical lines indicate the corresponding analytical lower and upper bounds (given by $P(x, y) \leq P(y_x) \leq P(x, y) + P(x')$ and $0 \leq P(y_x, y'_{x'}) \leq P(x, y) + P(x', y')$).

*set of SCMs that generate $P_M$.*

With sufficient PFN capacity, training drives the PFN predictive distribution to the exact Bayesian posterior over the causal query. Asymptotically, all remaining uncertainty is precisely partial-identification uncertainty and the support of the predictive distribution converges to the identified set $\mathcal{I}_\Psi(P_M)$.

**Theorem 2** (Frequentist Coverage). *Let $\mathcal{C}_{q_\omega(\Psi|\mathcal{D}_n), 1-\alpha}$ be a $(1 - \alpha)$ Bayesian credible set derived from the PFN $q_\omega(\Psi \mid \mathcal{D}_n)$. Then, the credible set $\mathcal{C}_{q_\omega(\Psi|\mathcal{D}_n), 1-\alpha}$ is an asymptotically valid frequentist confidence set for the identified set:*

$$\lim_{n \to \infty} P_M \left( \mathcal{I} \subseteq \mathcal{C}_{q_\omega(\Psi|\mathcal{D}_n), 1-\alpha} \right) = 1 - \alpha. \quad (8)$$

This result extends existing Bayesian-frequentist equivalence results for partially identified models. It implies that the Bayesian credible set for the identified set obtained in Thm. 1 achieves asymptotic frequentist coverage.

The following corollary summarizes the main contribution of the paper.

**Corollary 1** (Universality). *For any observational dataset $\mathcal{D}_n$ and counterfactual query $\Psi$, the $100\%$ Bayesian credible interval induced by the PFN $q_\omega(\Psi \mid \mathcal{D}_n)$, trained on a canonical prior, converges asymptotically to the true identified set:*

$$\mathcal{C}_{q_\omega(\Psi|\mathcal{D}_n), 1} \xrightarrow{n \to \infty} \mathcal{I}_\Psi(P_M). \quad (9)$$

Cor. 1 makes precise the claim that the proposed foundation model is a *universal* solver for partial identification.

## 4. Experiments

We demonstrate the use of causal foundation models (CFMs) for partial identification on binary two-variable systems for which analytical bounds are known. Specifically, we consider the inference of $P(y_x)$ and $P(y_x, y'_{x'})$ given samples from the observational distribution $P(x, y)$. For illustration,

| Model | $n$ | Cov. | Width | Inf. time (ms) |
|-------|-----|------|-------|----------------|
| Gibbs | 10 | 1.00 | 0.92 | 1053.2 |
| | 50 | 1.00 | 0.77 | 1095.2 |
| | 100 | 1.00 | 0.71 | 1146.5 |
| | 200 | 1.00 | 0.67 | 1322.2 |
| | 400 | 1.00 | 0.63 | 1521.1 |
| **CFM** | 10 | 0.99 | 0.70 | 4.2 |
| | 50 | 1.00 | 0.67 | 7.0 |
| | 100 | 1.00 | 0.67 | 8.9 |
| | 200 | 1.00 | 0.64 | 14.5 |
| | 400 | 1.00 | 0.63 | 24.9 |

*Table 1.* Coverage, mean interval width, and end-to-end inference time across context sizes for all evaluated models, averaged over 100 runs.

we plot in Fig. 1 posterior distributions from a randomly chosen $P(x, y)$, together with the corresponding tight analytical bounds derived by Manski (1990) and Tian & Pearl (2000).

In Tab. 1, we systematically compute the coverage, average width of derived intervals for all possible interventional and counterfactual queries, and inference time as a function of the number of context samples $n$, for the proposed method (CFM) as well as the Gibbs sampler proposed by (Zhang et al., 2022). Given that the Gibbs sampler requires a causal diagram as an input we take the union of the predicted sets under all possible causal diagrams of two variables.

## 5. Conclusion

We introduced a causal foundation-model approach to partial identification, where a Prior-Data Fitted Network is trained under a canonical prior with full support over discrete structural causal models. This formulation interprets counterfactual bounding as amortized posterior prediction over causal queries, with the potential to enable a single model to handle broad classes of queries, datasets, and structural assumptions without deriving problem-specific bounds each time.

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

# A. Related Work

In their seminal work in the early 1990's, Manski (1990); Robins et al. (2000) showed that the effect of an intervention may always be bounded in a non-trivial interval (*i.e.*, probabilities strictly contained in $[0, 1]$), irrespective of unobserved confounders or the causal structure underlying the variables involved. Causal effects are therefore in general said to be *partially identifiable*, *i.e.*, one can derive bounds that shrink the range of a priori possible values for a given effect and potentially serve as a useful support for decision-making.

Starting from this insight, the problem of partial identification has been attracting growing attention in the literature. To improve upon these bounds, one common restriction on the data generating mechanism is to assume knowledge of a causal graph describing the phenomenon of interest. Several results have derived progressively tighter bounds by exploiting the independencies in observational and interventional distributions implied by the causal graph (Balke & Pearl, 1997; Bellot, 2023; Sachs et al., 2026; Zhang, 2020; Zhang & Bareinboim, 2021a; Zhang et al., 2022; Bellot & Chiappa, 2024). For instance, Balke and Pearl (Balke & Pearl, 1997) (and subsequent refinements, *e.g.*, (Zhang et al., 2022)) defined polynomial optimization programs to derive bounds that are provably optimal given a causal diagram. More recent proposals consider parameterizations in the space of linear combinations of a set of fixed basis functions (Padh et al., 2023) and neural networks (Balazadeh Meresht et al., 2022; Hu et al., 2021). Analytical bounds (rather than numerical approximations) have also been derived in selected settings, such as for the "instrumental variable" graph (Zhang & Bareinboim, 2021a) and discrete systems with general graphs or equivalence classes (Bellot, 2023; Zhang, 2020). These results are all based on the assumption that the causal diagram (or equivalence class) is known exactly. They might be applied to the data-driven (graph-free) problem of partial identification by considering the union of the predicted sets under all possible causal diagrams of two variables (which, however, is difficult to scale to larger systems in practice).

In parallel, a number of works have adopted sensitivity assumptions (as an alternative or in combination with a causal diagram) that quantify the degree of unobserved confounding through various data statistics, such as odds ratios, propensity scores, etc. A rich literature on sensitivity assumptions exists, including Tan's sensitivity model (Tan, 2006) and Rosenbaum's sensitivity model (Rosenbaum, 2010). These approaches start with estimators that optimize the average or conditional treatment effect bounds subject to constraints implied by the sensitivity model.

While some of these works provide general methods for bounding the value of causal effects and counterfactuals, the approach we take in this paper is distinct. Rather than deriving problem-specific bounds via optimization or sensitivity-constrained estimation for each new dataset and query, we amortize partial identification through a Prior-Data Fitted Network trained once on synthetic examples from a *canonical prior* with full support over all structural causal models with discrete observables (Zhang et al., 2022). This also distinguishes our approach from existing causal foundation models (Robertson et al., 2025; Balazadeh Meresht et al., 2025; Dhir et al., 2025; Reuter et al., 2026; Ma et al., 2025), which target point-identified causal effects or place priors over restricted families of SCMs that need not assign mass to every counterfactual value compatible with the data; by construction, the support of our posterior predictive distribution asymptotically recovers the identified set, yielding bounds that are both valid and tight in the large-sample limit.

# B. Proofs

**Prop. 1 restated.** The canonical prior $\Pi_0$ in Eq. (6) has full support over $\mathbb{M}$: for any SCM $M \in \mathbb{M}$ and any open neighborhood $\mathcal{U}$ of $M$ in $\mathbb{M}$, $\Pi_0(\mathcal{U}) > 0$.

*Proof.* Fix an arbitrary SCM $M \in \mathbb{M}$ and any open neighborhood $\mathcal{U}$ of $M$. Write $M = (\mathcal{G}, \boldsymbol{\theta})$, where $\mathcal{G}$ is the semi-Markov graph and $\boldsymbol{\theta}$ are canonical parameters. Under Eq. (6), the prior factorizes as

$$\Pi_0(\mathcal{G}, \boldsymbol{\theta}) = \Pi_0(\mathcal{G})\Pi_0(\boldsymbol{\theta} \mid \mathcal{G}).$$

First, $\Pi_0(\mathcal{G}) > 0$ for every semi-Markov graph $\mathcal{G}$ on $\boldsymbol{V}$. Indeed, the directed-edge and bidirected-edge parts are independent Bernoulli draws with probabilities $q_D, q_B \in (0, 1)$, so any finite edge pattern has strictly positive probability.

Second, conditional on $\mathcal{G}$, the parameter prior is

$$\Pi_0(\boldsymbol{\theta} \mid \mathcal{G}) = \prod_{U \in \boldsymbol{U}} \mathrm{Dir}(\boldsymbol{\theta}_U; \alpha\boldsymbol{1}), \qquad \alpha > 0.$$

Each Dirichlet density is strictly positive on the relative interior of its simplex, hence has full support on that simplex; therefore the product prior has full support on

$$\Theta_{\mathcal{G}} = \prod_{U \in \boldsymbol{U}} \Delta_{|\mathrm{supp}_U|-1}.$$

So every non-empty open set in $\Theta_{\mathcal{G}}$ has positive $\Pi_0(\cdot \mid \mathcal{G})$ mass.

By (Zhang et al., 2022), canonical SCMs are representation-complete for discrete observables, and the map from canonical parameters to SCM-induced distributions/queries is continuous (polynomial in $\boldsymbol{\theta}$). Hence any open neighborhood $\mathcal{U}$ of $M$ induces a non-empty open parameter neighborhood around $\boldsymbol{\theta}$ within the graph component $\mathcal{G}$, so

$$\Pi_0(\mathcal{U} \mid \mathcal{G}) > 0.$$

Combining with $\Pi_0(\mathcal{G}) > 0$ gives

$$\Pi_0(\mathcal{U}) \geq \Pi_0(\mathcal{G})\Pi_0(\mathcal{U} \mid \mathcal{G}) > 0.$$

Since $M$ and $\mathcal{U}$ were arbitrary, $\Pi_0$ has full support on $\mathbb{M}$. $\qquad\square$

**Thm. 1 restated.** Let $\mathcal{D}_n$ be a dataset of $n$ i.i.d. observations generated from an SCM $M$. As $n \to \infty$, the predictive distribution $q_\omega(\Psi \mid \mathcal{D}_n)$ of the trained PFN converges weakly to the conditional prior distribution of the causal effect $\Psi$ given the true observational distribution $P_M$:

$$q_\omega(\Psi \mid \mathcal{D}_n) \xrightarrow{w} \Pi_0(\Psi \mid P_M), \tag{10}$$

where $\Pi_0(\Psi \mid P_M)$ is the prior distribution restricted to the set of SCMs that generate the same observational distribution $P_M$.

*Proof.* The training objective for the PFN is the minimization of the expected KL divergence between the true posterior and the approximation:

$$\omega^* = \arg\min_\omega \ \mathbb{E}_{M \sim \Pi_0} \mathbb{E}_{\mathcal{D}_n \sim P_M} \big[\mathrm{KL}\big(P(\Psi \mid \mathcal{D}_n) \,\|\, q_\omega(\Psi \mid \mathcal{D}_n)\big)\big]. \tag{11}$$

As shown in Müller et al. (2022) (Corollary 1.2), for a model with universal approximation capacity, the global optimum $q_{\omega^*}$ satisfies $q_{\omega^*}(\Psi \mid \mathcal{D}_n) = P(\Psi \mid \mathcal{D}_n)$ for all $\mathcal{D}_n$ for which the posterior is defined. Thus, the consistency of the PFN reduces to the consistency of the Bayesian posterior itself.

As $n \to \infty$, we aim to show that once the empirical distribution of the data converges to the true observational distribution $P_M$, which is point-identified, the remaining uncertainty characterizes the partially-identified bounds for the counterfactual query exactly.

We start by applying Doob's theorem (Doob, 1949) to the point-identified observational distribution $\mu = \phi(M) = P_M$. Following the recent treatment presented in (Miller, 2018, Thms. 2.1 and 2.2), we verify a number of regularity conditions:

- *Polish parameter space.* With the joint prior, the full parameter space is the disjoint union $\bigsqcup_{\mathcal{G}} \{\mathcal{G}\} \times \Theta_{\mathcal{G}}$, where $\Theta_{\mathcal{G}} = \bigtimes_{U \in \boldsymbol{U}} \Delta_{|\Omega_U|-1}$ is the simplex product for a fixed diagram $\mathcal{G}$. Since $p = |\boldsymbol{V}|$ is fixed there are finitely many semi-Markov diagrams on $p$ nodes, each $\Theta_{\mathcal{G}}$ is a compact metric space, and a finite disjoint union of compact metric spaces is again compact and separable, hence a complete separable metric space.

- *Measurability of $M \mapsto P_M(A)$.* In the canonical SCM, $P_M(\boldsymbol{V})$ is a polynomial function of the latent probabilities $\{P(u)\}_{u \in \Omega_U}$. Polynomials are continuous, and continuous functions on Polish spaces are Borel measurable.

- *Identifiability of the observational distribution.* The joint distribution $\mu = P_M(\boldsymbol{V})$ is point-identified from the empirical frequencies in $\mathcal{D}_n$; the mapping $M \mapsto P_M$ is many-to-one (multiple SCMs may induce the same $\mu$), but $\mu$ itself is identifiable.

- $L^1$ *integrability.* The counterfactual query $\Psi_M = \sum_{\boldsymbol{u}} \mathbb{1}\{\boldsymbol{Y_x}(\boldsymbol{u}) = \boldsymbol{y_x}, \ldots, \boldsymbol{W_z}(\boldsymbol{u}) = \boldsymbol{w_z}\} P(\boldsymbol{u})$ is bounded in $[0, 1]$, hence $\Psi \in L^1(\Pi_0)$.

Since $\mu$ is point-identified, Doob's theorem guarantees that for $\Pi_0$-almost all $M$, the posterior over the observational distribution concentrates:

$$P(\mu \mid \mathcal{D}_n) \xrightarrow{w} \delta_{P_M} \quad \text{as } n \to \infty. \tag{12}$$

While the posterior over the observational distribution concentrates, the posterior over the counterfactual query does not. To see this more precisely, we decompose the prior via the Disintegration Theorem. Let $\phi : \mathbb{M} \to \Delta(\boldsymbol{V})$ map each SCM to its observational distribution. The prior $\Pi_0$ decomposes as:

$$d\Pi_0(M) = d\Pi_0(M \mid \mu) \, d\Pi_\mu(\mu), \tag{13}$$

where $\Pi_\mu$ is the marginal over observational distributions and $\{\Pi_0(\cdot \mid \mu)\}_\mu$ is the family of conditional priors on the set of SCMs $\phi^{-1}(\mu)$. The Disintegration Theorem guarantees that this decomposition is unique and that the map $\mu \mapsto \Pi_0(\cdot \mid \mu)$ is measurable.

Let $h : \mathbb{R} \to \mathbb{R}$ be any bounded continuous test function. By the tower property:

$$\mathbb{E}[h(\Psi) \mid \mathcal{D}_n] = \int h(\Psi) \, d\Pi_0(M \mid \mathcal{D}_n) \tag{14}$$

$$= \int_\mu \left( \int_{M \in \phi^{-1}(\mu)} h(\Psi) \, d\Pi_0(M \mid \mu, \mathcal{D}_n) \right) d\Pi_0(\mu \mid \mathcal{D}_n) \tag{15}$$

$$= \int_\mu \underbrace{\left( \int_{\phi^{-1}(\mu)} h(\Psi(M)) \, d\Pi_0(M \mid \mu) \right)}_{=:H(\mu)} dP(\mu \mid \mathcal{D}_n). \tag{16}$$

The second equality follows from the Disintegration Theorem. For the third equality, in our setup the likelihood factorizes through the observational distribution $\mu$ : $L(\mathcal{D}_n \mid M) = L(\mathcal{D}_n \mid \mu)$. So if $M_1, M_2 \in \phi^{-1}(\mu)$, then $L(\mathcal{D}_n \mid M_1) = L(\mathcal{D}_n \mid M_2) = L(\mathcal{D}_n \mid \mu)$. Bayes' rule updates relative weights by prior times likelihood. If likelihood is identical for all $M \in \phi^{-1}(\mu)$, relative weights stay exactly as in the prior:

$$\Pi_0(M \mid \mu, \mathcal{D}_n) \propto \Pi_0(M \mid \mu) \times L(\mathcal{D}_n \mid \mu). \tag{17}$$

The factor $L(\mathcal{D}_n \mid \mu)$ is constant in $M$, so it cancels in normalization. Hence

$$\Pi_0(M \mid \mu, \mathcal{D}_n) = \Pi_0(M \mid \mu), \quad \text{for all } n. \tag{18}$$

Finally, since all observational and counterfactual quantities are polynomial functions of $\theta \in \Theta$ and $\Psi(M)$ is continuous, $H(\mu)$ is a continuous function of $\mu$. Applying the Continuous Mapping Theorem to the observational distribution (here denoted $P_{M^*}$) concentration result:

$$\int H(\mu) \, dP(\mu \mid \mathcal{D}_n) \longrightarrow H(P_{M^*}) = \int_{\phi^{-1}(P_{M^*})} h(\Psi(M)) \, d\Pi_0(M \mid P_{M^*}). \tag{19}$$

Since this convergence holds for every bounded continuous $h$, the Portmanteau theorem implies:

$$P(\Psi \mid \mathcal{D}_n) \xrightarrow{w} \Pi_0(\Psi \mid P_{M^*}). \tag{20}$$

Combining this with the optimality guarantee of the PFN, we conclude $q_\omega(\Psi \mid \mathcal{D}_n) \xrightarrow{w} \Pi_0(\Psi \mid P_{M^*})$.

$\square$

**Thm. 2 restated.** Let $\mathcal{C}_{q_\omega(\Psi \mid \mathcal{D}_n), 1-\alpha}$ be a $(1 - \alpha)$ Bayesian credible set derived from the PFN $q_\omega(\Psi \mid \mathcal{D}_n)$. Then, the credible set $\mathcal{C}_{q_\omega(\Psi \mid \mathcal{D}_n), 1-\alpha}$ is an asymptotically valid frequentist confidence set for the identified set:

$$\lim_{n \to \infty} P_M\big( \mathcal{I} \subseteq \mathcal{C}_{q_\omega(\Psi \mid \mathcal{D}_n), 1-\alpha} \big) = 1 - \alpha. \tag{21}$$

*Proof.* The proof is an application of Kline & Tamer (2016, Thm. 5) to our set-up. For this we verify the conditions of their theorem: Assumptions 1, 3, 5, 6 in (Kline & Tamer, 2016).

Let $\mu := P_M(\boldsymbol{V})$ be the so-called reduced-form parameter (cell probabilities of the observational law) in (Kline & Tamer, 2016), and define the identified-set map

$$\Gamma(\mu) := \mathcal{I}(\mu) = [\ell(\mu), r(\mu)],$$

where $\ell(\mu), r(\mu)$ are the optimal values of the optimization program in Eq. (2). Thus our model is exactly a map from a point-identified reduced form $\mu$ to an identified set for the partially identified target.

The parameter space for $\mu$ is a simplex (hence a Borel subset of a finite-dimensional Euclidean space), so Assumption 1 in Kline & Tamer (2016) holds. Since observations are i.i.d. categorical, the reduced-form posterior satisfies a Bernstein–von Mises limit and the frequentist CLT for $\hat{\mu}_n$ holds:

$$\sqrt{n}(\mu - \hat{\mu}_n) \mid \mathcal{D}_n \Rightarrow N(0, \Sigma_0), \qquad \sqrt{n}(\hat{\mu}_n - \mu^*) \Rightarrow N(0, \Sigma_0),$$

which match Assumptions 3 and 6 in Kline & Tamer (2016). For Assumption 5 (their asymptotic independence condition), we impose the standard smooth-interval sufficient condition from Kline & Tamer (2016, Remark 5 and Lemma 1): near $\mu^*$, $\Gamma(\mu)$ is nonempty and the endpoint maps $\ell(\mu), r(\mu)$ are regular enough for the (Bayesian and frequentist) delta method.

Let $\mathcal{I}_{\Pi_0(\Psi \mid \mathcal{D}_n), 1-\alpha}$ denote a $(1 - \alpha)$ Bayesian credible set for the identified set, i.e.

$$\Pi\big( \Gamma(\mu) \subseteq \mathcal{I}_{\Pi_0(\Psi \mid \mathcal{D}_n), 1-\alpha} \mid \mathcal{D}_n \big) = 1 - \alpha.$$

By Kline & Tamer (2016, Thm. 5),

$$P_{M^*}\big( \Gamma(\mu^*) \subseteq \mathcal{I}_{\Pi_0(\Psi \mid \mathcal{D}_n), 1-\alpha} \big) \to 1 - \alpha.$$

Since $\Gamma(\mu^*) = \mathcal{I}(P_{M^*})$, this is the desired frequentist coverage statement. Finally, because Thm. 1 gives $q_\omega(\Psi \mid \mathcal{D}_n) \Rightarrow \Pi_0(\Psi \mid P_{M^*})$ where $M^*$ is the true SCM: the PFN-based credible sets asymptotically coincide with the corresponding Bayesian credible sets for the identified set and inherit the same limit coverage. $\square$

**Remark.** If the causal effect happens to be point-identified (e.g., under unconfoundedness and known causal diagram), the identified set collapses to a point and Thm. 2 reduces to the standard frequentist coverage guarantee for a point-identified parameter.

**Remark.** Compared with recent amortized causal PFN theory (Balazadeh Meresht et al., 2025; Ma et al., 2025), our guarantee targets a different object: under partial identification, the PFN predictive converges weakly to the *conditional prior* $\Pi_0(\Psi \mid P_M)$ on the collection of SCMs compatible with the true observational law, whereas they study identified causal functionals and aim for point recovery. The proof strategy is similar: all use Doob-type concentration on the observational parameter and PFN optimality (Müller et al., 2022) to reduce the network to the exact Bayesian predictive; though an important difference is that Balazadeh Meresht et al. and Ma et al. impose identification restrictions on the prior and assume that the map $M \mapsto P_M$ is measurable, we explicitly prove and verify that this is true under the canonical SCM prior for discrete systems that we propose.

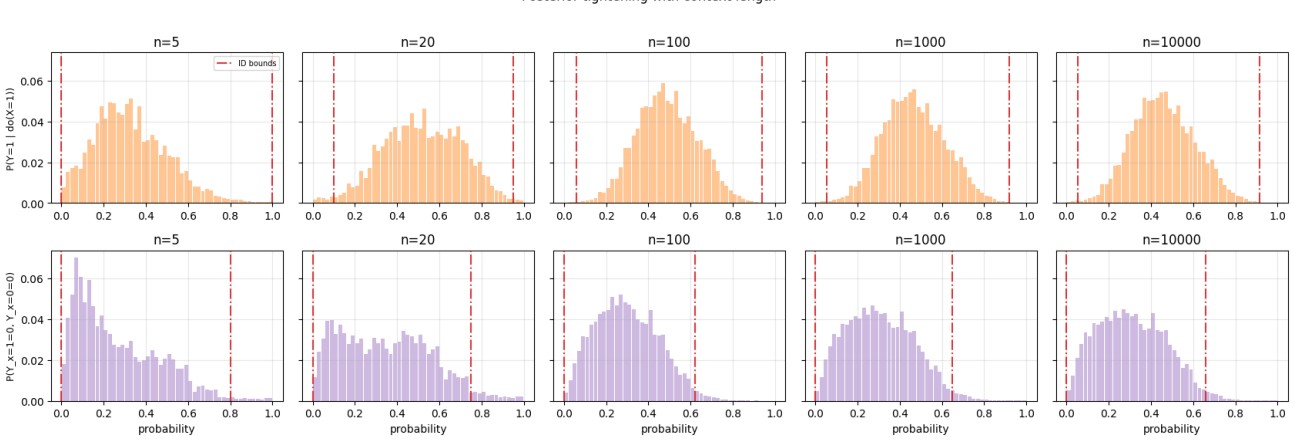

*Figure 2.* Predicted posterior for the interventional query $P(Y_{x=1} = 1)$ and the counterfactual query $P(Y_{x=1} = 0, Y_{x=0} = 0)$ as a function of the number of context samples $n$ for a randomly drawn SCM.

## C. Additional Experiments

In this section, we consider additional experiments to evaluate the performance of the CFM.

We start by analyzing the predicted posterior distributions for the interventional and counterfactual queries as a function of the number of context samples $n$ for a randomly drawn SCM. Fig. 2 shows the predicted posterior distributions for the interventional query $P(Y_{x=1} = 1)$ and the counterfactual query $P(Y_{x=1} = 0, Y_{x=0} = 0)$ as a function of the number of context samples $n$. We see that the posterior distributions approximately concentrate on the identified set as the number of context samples increases.

We also evaluate the performance of the CFM on a three variable system with binary variables $X, Y, Z$ given samples from the observational distribution $P(x, y, z)$. To make comparisons with existing partial identification methods, we will restrict the prior to sampling SCMs with a fixed causal diagram $\{Z \to X \to Y, X \leftrightarrow Y\}$: the instrumental variable graph. We generate $n$ samples from a randomly chosen $P(x, y, z)$ and compute the coverage, average width of the derived interval, and inference time as a function of the number of context samples $n$. We evaluate 95 percent credible intervals. The results are shown in Tab. 2.

| Model | $n$ | Cov. | Width | Inf. time (ms) |
|---|---|---|---|---|
| Gibbs | 10 | 1.000 | 0.854757 | 822.981473 |
| | 20 | 1.000 | 0.790407 | 839.420002 |
| | 50 | 1.000 | 0.706258 | 842.181667 |
| | 100 | 1.000 | 0.660208 | 822.117917 |
| | 200 | 1.000 | 0.623506 | 951.234029 |
| | 400 | 1.000 | 0.601055 | 1062.213033 |
| **CFM** | 10 | 0.900 | 0.549325 | 2.481694 |
| | 20 | 0.950 | 0.535600 | 2.994190 |
| | 50 | 0.950 | 0.471782 | 4.435860 |
| | 100 | 0.975 | 0.471999 | 5.642881 |
| | 200 | 0.970 | 0.472965 | 9.045406 |
| | 400 | 0.985 | 0.467423 | 17.795929 |

*Table 2.* Coverage, mean interval width, and end-to-end inference time across context sizes for all evaluated models, averaged over 100 runs.

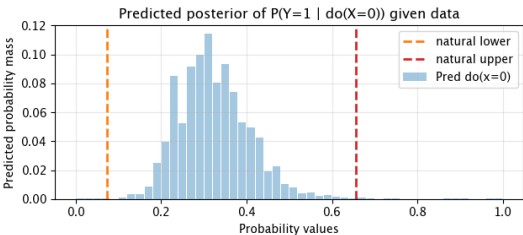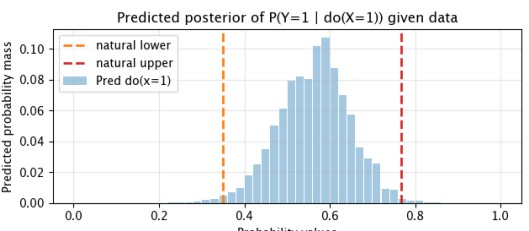

*Figure 3.* Predicted posterior distributions for $\Psi := P(Y_x = 1)$ at $x = 0$ (left) and $x = 1$ (right) in the three-variable binary SCM experiment. Dashed vertical lines indicate the corresponding analytical lower and upper bounds (Balke & Pearl, 1994).

## D. Architecture and Training Details

This appendix gives a complete description of the neural network architecture and training procedure used in the experiments.

### D.1. Architecture

The model $q_\omega(\Psi \mid \mathcal{D})$ is implemented as a Neural Process composed of three components: a context/query encoder, an outcome predictor head, and a loss function. A single forward path handles both interventional and counterfactual queries via a unified query specification.

**Embedding.** Each training episode pairs the shared context $\mathcal{D}$ with a batch of $N_q$ queries. Query $q$ targets a counterfactual probability $\Psi_q = P_M(\boldsymbol{y_x}, \ldots, \boldsymbol{z_w})$ that is represented as a conjunction of potential-outcome events (sharing the same latent draw $\boldsymbol{u} \sim P(\boldsymbol{u})$). A counterfactual probability is encoded as a 2-d array capturing the index of the node that is intervened upon, its value, and the index and value of the outcome node. For example, $P(Y_{X=1} = 1, Y_{X=0} = 0)$ in a two variable system is encoded as:

| event | int. node | int. val | out. node | out. val |
|-------|-----------|----------|-----------|----------|
| 0 | 0 $(x)$ | 1 | 1 $(y)$ | 1 |
| 1 | 0 $(x)$ | 0 | 1 $(y)$ | 0 |

For queries with less than a pre-specified maximum number of events $K$, the remaining events are padded.

Each query is embedded into a token of shape [num nodes, model dimension]. Each node embedding varies depending on the role (intervention or outcome variable), value, and event index, that is assigned to a node in the query. Each type of embedding is learned separately and the final embedding is the sum of the contributions from all the nodes in the query. For example, for the query $P(Y_{X=1} = 1, Y_{X=0} = 0)$ in a two variable system, the embedding of node $X$ is given by the sum of the intervention embedding for $X = 1$ in the first event and the outcome embedding for $X = 0$ in the second event.

The benefit of this approach is that a query node $j$ attends to context node $j$, giving a clear inductive bias for reading evidence about the variables involved in the query, and is efficient as complex queries can be summarized in a single token while keeping track of the events they are involved in. However, it does have the drawback of summing multiple events into the same node slots, which can blur distinctions when events overlap on the same variables.

**Encoder.** The encoder concatenates context and query tokens along the sample axis and processes a batch through $L$ alternating attention blocks. Each block applies two sub-layers in sequence. Sample attention operates across tokens for each node independently, with a causal mask that enforces query tokens attend only to context tokens (and not to each other); Node attention operates across nodes within each token. Both sub-layers use pre-norm residual connections (LayerNorm $\rightarrow$ MHA $\rightarrow$ add) followed by a position-wise feed-forward network with a SwiGLU activation. The feed-forward hidden dimension is $4d$. Dropout is applied after each activation. After $L$ such blocks, only the query slice of the output is retained, yielding a representation tensor of shape $[B, N_q, \text{num nodes}, \text{model dimension}]$.

### D.2. Loss

Two quantities are extracted from the encoder output for each query: an outcome and an intervention summary. The outcome representation is obtained by averaging the representation vectors at the outcome-node positions across the query's $K$ active

worlds. The intervention summary is the mean intervention value across events. These are concatenated to form a predictor input which is fed to a two-layer MLP with SwiGLU activations followed by a linear projection to $B$ logits. The $B$ bins partition $[0, 1]$ into equal-width intervals so that the predicted probability of each bin defines a histogram approximation to the predictive distribution over $\Psi \in [0, 1]$. The model's point prediction is the expectation under this histogram.

Given a training example $(\mathcal{D}, \{\Psi_q\}_{q=1}^{N_q})$ where each $\Psi_q \in [0, 1]$ is the ground-truth probability of the corresponding query, the per-query training loss is a cross-entropy over bins:

$$\ell(\hat{p}_q, \Psi_q) = -\log \hat{p}_{q,b_q^*}, \qquad b_q^* = \min(\lfloor \Psi_q \cdot B \rfloor, B - 1).$$

This corresponds to assigning the ground-truth probability $\Psi_q$ to its enclosing bin and maximising the predicted mass on that bin. The loss is averaged over all valid queries in the batch.

### D.3. Training Procedure

Each training step draws a fresh batch of episodes. An episode consists of a context set of $N_c$ i.i.d. observations sampled from the observational distribution $P_M(\mathcal{V})$ of a randomly drawn canonical SCM $M \sim \Pi_0$, together with $N_q$ structured queries against that shared context, each paired with its ground-truth counterfactual probability $\Psi_q(M) = P_M(\boldsymbol{y_x}, \ldots, \boldsymbol{z_w})$. Context size $N_c$ is sampled uniformly at random from $[N_{\min}, N_{\max}]$ at each step, exposing the model to varying amounts of observational evidence during training.

Table 3 summarises the hyperparameters used in the experiments.

*Table 3.* Model and training hyperparameters.

| Parameter | Value | Description |
|---|---|---|
| $d$ | 64 | Model (embedding) dimension |
| $L$ | 3 | Number of alternating attention blocks |
| $H$ | 4 | Number of attention heads |
| $B$ | 50 | Number of histogram bins |
| MLP depth | 2 | Depth of predictor MLP |
| FF dim | $4d = 256$ | Feed-forward hidden dimension |
| Dropout | 0.0 | Dropout probability |
| Batch size | 16 | Episodes per gradient step |
| $N_q$ | 250 | Queries per episode |
| $W_{\max}$ | 2 | Maximum counterfactual atoms (worlds) per query |
| $N_{\min}$ | 5 | Minimum context size |
| $N_{\max}$ | 5000 | Maximum context size |
| Steps | 3000 | Total training steps |
| Learning rate | $10^{-3}$ | Peak learning rate (AdamW) |
| Weight decay | 0 | AdamW weight decay |
| Warmup fraction | 0.2 | Fraction of steps for linear warmup |
| LR schedule | cosine | Cosine decay after warmup |
| Gradient clip | 1.0 | Max gradient norm |

