# OpenReview forum: "Foundation Models for Partial Causal Identification"
_ICML.cc/2026/Workshop/FMSD — FMSD @ ICML 2026 Poster_

### Official Review · Reviewer_qdQc · 2026-05-13
**Highly original and significant contribution**

**Rating:** 9
**Confidence:** 4

**Review:**

This paper presents a highly original and significant contribution to causal inference by extending the foundation model paradigm to partial identification, a notoriously fragmented field where bounds usually require bespoke mathematical derivations. The work demonstrates high theoretical quality and clarity, rigorously defining a "canonical prior" and proving that pairing it with Prior-Data Fitted Networks (PFNs) yields valid asymptotic bounds and frequentist coverage. The pros of this approach include a massive reduction in inference time compared to traditional Gibbs sampling (e.g., dropping from over 1000ms to under 25ms), strong mathematical guarantees for consistency, and a generalized framework capable of handling varied causal queries without new problem-specific derivations. However, the cons involve a strict limitation to structural causal models with discrete, finite observable variables, an inherent reliance on the PFN's universal approximation capacity to actually achieve the optimal theoretical posterior, and empirical validation that is currently restricted to very small-scale toy systems involving only two to three variables.

---

### Official Review · Reviewer_f8Hd · 2026-05-20
**Prior Fitted Network for Partial Identification**

**Rating:** 6
**Confidence:** 5

**Review:**

# Summary
This paper proposes a causal foundation model for partial causal identification. The idea is to extend the CausalPFN paradigm from point estimation of causal effects to settings where the causal query is not identifiable due to hidden confounding. The model is trained on synthetic SCMs sampled from a canonical full-support prior over discrete causal models, including bidirectional edges representing unobserved confounding. Given the observational data, the model is able to predict a binned posterior distribution and the distribution is interpreted as an estimate of the identified set.

# Strengths
1. The paper tackles a real limitation of current causal foundation models: causal effects are not point-identifiable under hidden confounding.
2. The canonical SCM prior is well motivated and gives a good argument why the posterior support can recover the partial-identification interval.

# Areas for Improvement
1. The empirical comparison is limited. The method is very close to CausalPFN, but the paper does not compare against CausalPFN baselines. (A fixed uncertainty can be added to CausalPFN to allow interval prediction)
2. The paper should ablate the SCM choices. Canonical prior vs simpler priors, hidden-confounding prior vs no hidden confounding, and binned posterior prediction vs quantile prediction.
3. The experiments are small, mostly two- and three-variable binary SCMs.

# Detailed Comments
A key missing baseline is a CausalPFN point predictor with calibrated intervals using a validation set

# Justification of Score
I would rate this as weak accept. The idea is interesting although the empirical evidence is not yet sufficient enough.

---

### Official Review · Reviewer_yeQd · 2026-05-20
**A causal foundation model for partial identification, integrity concerns**

**Rating:** 3
**Confidence:** 3

**Review:**

# Summary
This paper proposes a causal foundation model for partial identification. The key contribution is a canonical prior over discrete SCMs with full support, which guarantees that the posterior support asymptotically captures the identified set for any counterfactual query. A PFN is trained under this prior to amortize inference over causal queries from observational data.

# Strengths
The paper cleanly connects partial identification with the PFN / causal foundation model paradigm

Attempting to build a unified framework for arbitrary counterfactual queries under partial identification is ambitious and intellectually interesting

# Weaknesses
The theoretical guarantees rely on strong asymptotic and regularity assumptions, thus it is unclear to me how informative these guarantees are in practical finite-sample settings

The paper claims to develop a universal solver for partial identification but the experimental comparison is limited to a Gibbs sampler baseline. The authors cite partial identification methods (Duarte et al., 2024, Sachs et al., 2023) but do not compare against them. While it is possible that this comparison would not be possible to run, that issue is not discussed. Otherwise, without such comparisons it is impossible to assess whether the proposed approach offers practical advantages over existing tools.

Architecture and training setup details are missing, which hurts reproducibility.

Experiments are limited to two and three variable SCMs. It's not clear if the approach scales to graphs of practical size.


# Integrity Concern
The reference list contains at least one citation that does not correspond to any existing publication:

Kline, B. and Tamer, E. "Bayesian inference in partially identified models: Is the shape of the prior what matters?" Journal of Econometrics, 190(2):328–346, 2016.

The actual Kline & Tamer (2016) paper is titled "Bayesian inference in a class of partially identified models," published in Quantitative Economics, 7(2):329–366. The cited title closely mirrors a separate paper by a different author: Gustafson (2014), "Bayesian inference in partially identified models: Is the shape of the posterior distribution useful?", Electronic Journal of Statistics. The cited entry appears to be a composite of these two real papers. It has correct authors and year from one, title structure from the other, but neither journal nor page numbers match any existing work.

(minor) Additionally, Dhir et al. is listed in the references as "Dhir, A. et al.". Using "et al." in a reference list entry rather than enumerating co-authors is not standard practice and is consistent with automated reference generation without verification. The full author list is Dhir, Anish, Cristiana Diaconu, Valentinian Lungu, James Requeima, Richard Turner, and Mark van der Wilk.

These issues raise concerns about whether the reference list was generated / completed with an AI language model without adequate verification.